# *Spergularia hanoverensis* (*Caryophyllaceae*): Validation and Recircumscription of a Misinterpreted Species from South Africa

**DOI:** 10.3390/plants12132481

**Published:** 2023-06-28

**Authors:** María Ángeles Alonso, Manuel B. Crespo, Mario Martínez-Azorín, Ladislav Mucina

**Affiliations:** 1Departamento de Ciencias Ambientales y Recursos Naturales (dCARN), Universidad de Alicante, P.O. Box 99, ES-03080 Alicante, Spain; crespo@ua.es (M.B.C.); mmartinez@ua.es (M.M.-A.); 2Iluka Chair in Vegetation Science and Biogeography, Harry Butler Institute, Murdoch University, 90 South Street, Murdoch, Perth, WA 6150, Australia; ladislav.mucina@murdoch.edu.au; 3Department of Geography and Environmental Studies, Stellenbosch University, Private Bag X1, Matieland, Stellenbosch 7602, South Africa

**Keywords:** ITS phylogeny, plant endemics, plant morphology, South African flora, *Spergularia*, *Sperguleae*, taxonomy, *trnL-trnF* phylogeny

## Abstract

The name “*Spergularia hanoverensis* Simon” has been misapplied to an endemic taxon confined to inland semidesert ecosystems in central-western South Africa. It is commonly accepted as a small annual species occurring in saline habitats in a wide elevation range, but its identity still remains obscure. In the context of taxonomic and phylogenetic research on the African species of *Spergularia*, we found that the name was never validly published. After revision of herbarium material housed in South African herbaria, a voucher collected from Hanover was found at PRE bearing some labels handwritten by E. Simon that suggest it might be an intended type for the name. Additional herbarium material and wild populations from the Karoo region were identified that matched the samples in that voucher, and taxonomic research was conducted to clarify their identity. Among other characters, those Karoo plants show a woody dense compact habit, woody perennial at base; stems prostrate to ascendent; leaves entirely glabrous, somewhat glaucous; large white-hyaline conspicuous stipules; inflorescence glanduliferous, many-flowered subdichasial cyme, with minute bracts; flowers small, with white petals approximately equalling sepals in length, stamens 7–8, and styles free from base; capsule small, with seeds dimorphic, unwinged to broadly winged, with testa always densely tuberculate. Molecular analyses of plastid (*trnL*-*trnF* region) and nuclear ribosomal (5.8S-ITS2 region) DNA sequence data support morphological differentiation of the Karoo plants, for which the name *S. hanoverensis* is here effectively published. A full morphological description and data on ecology, habitat, distribution, and taxonomic and phylogenetic relationships of *S. hanoverensis* are compared to other members of the “South African group”, namely *S. glandulosa*, *S. namaquensis*, and *S. quartzicola*, from which the new species considerably differs. The adaptative significance of dimorphic seeds of *S. hanoverensis* is briefly commented on in the context of the species habitat preference. An identification key is presented for the South African related taxa.

## 1. Introduction

*Spergularia* (Pers.) J.Presl & C.Presl, nom. cons., comprises ca. 60 species of both annual and perennial plants with worldwide distribution, but mainly occurring in the Mediterranean Basin and temperate South America [1]. Some members of the genus are narrow endemics, while others are subcosmopolitan synanthropic plants [2]. In most cases, they occur on soils rich in mineral salts, such as chlorides, sulphates, and nitrates, in both natural and human-disturbed habitats.

Recently, molecular phylogenies recovered a consistent clade, including *Spergularia*, *Spergula* L., and *Rhodalsine* Gay. Some authors [3,4] considered this group a distinct tribe to which the name *Sperguleae* Dumort. was applied, thus disregarding the traditional tripartite subfamilial arrangement of *Caryophyllaceae* Juss. [5], in which *Spergularia* was placed in tribe *Polycarpaeae* DC., subf. *Paronychioideae* (Juss.) Meissn. [2,5].

In the last decades, the relationships of *Spergularia* to other caryophyllid genera, such as *Spergula*, have been controversial [6]. However, recent phylogenies by Kool and Thulin [7], based on plastid DNA sequences, recovered samples of *Spergula* and *Spergularia* (including *Sanctambrosia* Skottsb. and *Spergularia flaccida* (Madden) I.M.Turner = *S. fallax* Lowe) forming monophyletic sister clades, supported by the different number of styles and fruit valves, in both cases 5 vs. 3, respectively [8,9]. In this context, maintaining the separate identities of both genera appears to be a good choice, which is followed here.

Regarding the *Spergularia* representatives in South Africa, Alonso et al. [8] have recently reinstated or validated two neglected species from western South Africa, i.e., *S. glandulosa* (Jacq.) Heynh. and *S. namaquensis* Schltr. ex M.Á.Alonso et al., and have also described [9] a rare endemic restricted to the quartz outcrops from Knersvlakte (SW Namaqualand), i.e., *S. quartzicola* M.Á. Alonso et al. All three species are perennials that were usually misidentified as the subcosmopolitan halophytic *S. media* (L.) C.Presl or as the nitrophilous alien *S. bocconei* (Scheele) Graebn. However, sound morphological vegetative and reproductive differences allow easy distinction from the rest of the members of the genus [9]. Those three outstanding newly recovered or described taxa are to be added to the other species of *Spergularia* usually referred to in South Africa [10,11,12]. One of those is the enigmatic *S. hanoverensis* E.Simon, a name never validly published [13].

“*Spergularia hanoverensis* Simon” was reported for the first time by Olivier and Germishuizen [14] in the list of South African *Spergularia*, along with seven other taxa (six species plus a variety) for which no additional data were supplied. The species was not mentioned by Goldblatt and Manning [10], who only accepted three species in that genus, namely *S. media*, *S. pallida* G.Don (erroneously cited as “*Spergularia pallida* (Dumort.) Piré”, the authorship that corresponds to *Stellaria pallida*), and *S. rubra*. The name *Spergularia pallida* is currently treated in the synonymy of *S. media*. Later, Nkonki [15], in her compilation of the South African *Caryophyllaceae*, recovered *S. hanoverensis* and listed it together with three representatives of that genus: *S. bocconei*, *S. media*, and *S. rubra*. There, *S. hanoverensis* was catalogued as “Annual. Herb. Ht?—0.03 m. Alt 300–1370 m. NC, WC”, thus indicating that it was considered to be an annual herb, rather small in size, and occurring within a broad elevation range through the Northern Cape and Western Cape provinces of South Africa. However, that name was not listed by Snijman [12], who again only recognised three species (*S. bocconei, S. media*, and *S. rubra*) in the Extra Cape Subregion (Greater Cape Floristic Region) of South Africa. In any case, the current application of the name *S. hanoverensis* in Nkonki’s sense [15] often does not fit with the scarce herbarium material so identified, and hence, its identity remains confusing.

As part of an investigation project of the “H2020 Research and Innovation Staff Exchange (RISE) Programme” of the European Commission (project num. 645636), intensive field work was carried out in western South Africa between 2015 and 2018, which has been complemented in subsequent years with new field explorations. During that research, some populations of an outstanding subshrubby and strongly woody at base, glabrous, white-flowered member of *Spergularia* were observed in inland saline and subsaline habitats of the Nama-Karoo biome, mostly in the Northern Cape province. Closely examining living material and vouchers from South African herbaria revealed that those Karoo populations exhibited a combination of macro- and micromorphological characters and ecology absent in any described South African taxa of the genus. However, they are a perfect match with samples in a voucher labelled “*Spergularia hanoverensis*” made by the French botanist Eugène E. Simon, which is conserved in the Pretoria herbarium (PRE). This material is akin to other members in the so-called “South African taxa” group of *Spergularia* (sensu [8]), which sometimes were misidentified as *S. media*. Phylogenetic studies of the Karoo plants, using nuclear (5.8S-ITS2 region) and plastid (*trnL*-*trnF* region) DNA sequence data, revealed that they could not be identified either with the Northern Hemisphere taxa of the *S. media* group nor with other South African members of the genus.

In the present contribution, *Spergularia hanoverensis* is validly described for a plant endemic to the Nama-Karoo (NK) biome [16] in western South Africa, according to the original concept “in schedis” of E.E. Simon. A brief story on that name is presented, and data on its morphology, ecology, distribution, and phylogenetic relationships are reported that support recognition of *S. hanoverensis* at specific rank in the “South African taxa” group.

## 2. Materials and Methods

### 2.1. Morphological and Habitat Studies

Detailed morphological studies were undertaken on both living plants from wild populations and herbarium specimens sourced from the herbaria ABH, BOL, GRA, HBG, K, M, NBG, P, and PRE (acronyms according to Thiers [17]), using an OLYMPUS SZX7 binocular microscope. A personal collection of one of the authors (L.M.) that is currently deposited at ABH, including numerous vouchers of *Spergularia*, was also studied. Thirty living individuals from two wild populations of *S. hanoverensis* in Northern Cape province (i.e., Karreekop farm, between Brandvlei and Williston, and Zoekop Farm, SW of Middelpos, on the road to Ganaga Pass) were sampled and analysed in situ. Further, thirteen herbarium vouchers of *S. hanoverensis* were studied (see Section 3.2), some of them including several duplicate sheets, and with several complete individuals each. The obtained morphological data were focussed on sufficiently illustrating the intraspecific variation of the newly described species, concerning other related perennial South African taxa of the genus, as summarised by Alonso et al. [8,9]. Digital images of *Spergularia* from iNaturalist (https://www.inaturalist.org/observations/?place_id=any&taxon_id=58170; accessed on 2 May 2023) were also checked, and some were found to meet the distribution of the studied species.

Scanning electron microscope (SEM) micrographs of seeds were taken with a JEOL JSM-IT500HR operating at 15 kV. No special treatment of the material was required prior to observation. At least 5 mature seeds from different individuals of *S. hanoverensis* from different sites (Table 1) were observed in detail when available. Samples were glued directly onto metallic stubs and then coated with 10 nm platinum in a QUORUM Q150T ES Plus sputter coater. The ImageJ software [18] was used for measurements on SEM micrographs.

Authors of the taxa cited in the text follow IPNI [19]. Nomenclatural issues accord with Turland et al. [20]; orthography of geographical names agrees with Leistner and Morris [21]; and the grid-number system is in accordance with the National Geospatial Information [22]. Bioclimate, bioregion, and vegetation classification agree with Mucina and Rutherford [16].

### 2.2. Molecular Analyses

Herbarium vouchers and silica-gel-dried material were used for total DNA extraction employing a modified 2 × cetyltrimethylammonium bromide (CTAB) protocol [23]. For *Spergularia hanoverensis*, sampling from herbarium material was not permitted, and hence, only silica-gel-dried material from two wild populations (one sample per population) was utilised. Addition of further samples from those same populations did not modify the phylogenetic trees. Total DNA was purified using MOBIO minicolumns and kept in 0.1 × TE buffer (10 mM Tris-HCl, 1 mM ethylenediaminetetraacetic acid (EDTA), pH 8.0). The *trnL*-*trnF* region (hereafter *trnL*-*F*) of chloroplast DNA (cpDNA) was amplified using specific primers *trnL*-BOC and *trnL*-BOF-R as described in Oxelman et al. [24], whereas the whole internal transcribed spacer—ITS—region (ITS1 spacer, 5.8S gene, ITS2 spacer) of nuclear ribosomal DNA (nrDNA) was amplified using the ITS1 (forward) and ITS4 (reverse) primers [25] and then adjusted to match the length of the 5.8S-ITS2 (hereafter ITS2) sequences retrieved from GenBank for alignment. Amplifications were performed on a reaction volume of 25 μL containing 22 μL of ABGene 1.1 × Master Mix, 2.5 mM MgCl_2_ (Thermo Scientific, Waltham, MA, USA), 0.5 μL of 0.4% bovine serum albumin (BSA), 0.5 μL of dimethyl sulfoxide (DMSO), 0.5 μL of each primer (10 pmol/μL), and 1 μL of template DNA on a 9700 GeneAmpl thermocycler (Applied Biosystems). The PCR programme for *trnL*-*F* was as follows: 2 min at 97 °C, followed by 35 cycles of 97 °C for 20 s, 55 °C for 50 s, 72 °C for 1.5 min, and a final extension at 72 °C for 8 min. The PCR programme for ITS2 was as follows: 2 min at 95 °C, followed by 30 cycles of 95 °C for 1 min, 53 °C for 1 min, 72 °C for 2 min, and a final extension at 72 °C for 5 min.

Sequencer 4.1 (Gene Codes Corp., Ann Arbor, MI, USA) was used to assemble complementary strands and verify software base-calling. Sequence alignment was performed using MUSCLE [26] conducted in MEGA X v.10.2.6 [27] with minor manual adjustments to obtain the final aligned matrix. Forty-five samples belonging to twenty-four species of *Spergularia* were used for phylogenetic reconstructions, using *Rhodalsine geniculata* (Poir.) F.N.Williams, *Rh. platyphylla* Gay in Christ, *Spergula arvensis* L., *Spergula morisonii* Boreau, and *Spergula pentandra* L. as outgroups. Two datasets were built: one for the *trnL*-*F* region of cpDNA (matrix with 33 sequences and 990 positions) and another for the ITS2 region of nuclear nrDNA (matrix with 25 sequences and 282 positions). Sequences of each region were retrieved from different plant sources depending on GenBank availability [8,9], except for four of them gathered in South Africa and Spain, which were generated specifically for the present study (Table 2). For that reason, most accessions in the *trnL-F* and ITS2 datasets do not come from a unique plant source. Accessions from GenBank filed as “*Spergularia fallax* Lowe” are shown in our trees as *S. flaccida* (Madden) I.M.Turner, the name having priority for that species.

Phylogenetic analyses of both regions were obtained using maximum parsimony (MP), maximum likelihood (ML), and neighbour joining (NJ) methods. MP analysis was conducted in both PAUP (using heuristic search options with the tree searching strategy based on nearest neighbour interchange, NNI) and MEGA (using heuristic search options with the tree searching strategy based on subtree-pruning-regrafting—SPR—with search level 1; [28]) for result comparison, with 10,000 replicates. ML [29] and NJ [30] analyses were also performed in MEGA, as well as the selection of the best model of DNA substitutions for each method using the Akaike information criterion (AIC; [31]); models with the lowest BIC (Bayesian information criterion) scores were considered to best describe the substitution pattern for the ML and NJ analyses. Phylogenetic reconstructions for ML and evolutionary distances for NJ for the ITS2 matrix were estimated using the K2 model (2-parameter method of Kimura [32]) with the rate variation model allowing for some sites to be evolutionarily invariable (+*I*, 30.26% sites). In contrast, for the *trnL-F* matrix, the T92 model (3-parameter method of Tamura [33]) was applied, with a discrete gamma distribution (*G* = 0.677) modelling the rate variation among sites. In every case, all sites in the matrixes were considered. For comparison purposes, remotion of all ambiguous positions for each sequence pair (pairwise deletion option) was also performed, and no significant differences (only affecting BS values in a few branches) were observed in the obtained phylogenies. For all those methods, support was assessed by the bootstrap [34] with 10,000 replicates but holding only 10 trees per replicate. Clades showing bootstrap percentage (BP) values of 50–74% were considered weakly supported, 75–89% moderately supported, and 90–100% strongly supported.

Furthermore, Bayesian inference (BI) analyses were conducted with MrBayes 3.2 [35], in which the Markov chain Monte Carlo (MCMC) algorithm was run for 10 million generations and sampled every 1000 generations. The general time reversible (GTR) + proportion of invariant sites (*I*) + gamma distribution (*G*) model was used in the analyses (set nst = 6 rates = invgamma), according to the results obtained with jModelTest 2.1.10 [36] under AIC. The first 25% of generations (burninfrac = 0.25) were excluded, and the remaining trees were used to compile a posterior probability (PP) distribution using a 50% majority-rule consensus.

More detailed information on plant material sources, GenBank accessions, DNA extraction and sequencing, and data analyses are provided in Alonso et al. [8,9].

## 3. Results and Discussion

### 3.1. On the Trail of Spergularia hanoverensis: A Brief Story of the Name and Its Application

According to Nkonki [15], the name *S. hanoverensis* should be applied to annual plants occurring in broad elevation and geographical ranges through Northern and Western Cape. However, as previously mentioned, herbarium material labelled *S. hanoverensis* usually shows morphological traits not matching that broadly assumed concept (see below).

The recovery of this almost forgotten name most probably followed Nkonki’s revision of the *Spergularia* vouchers conserved in the National Herbarium at Pretoria. Among them, the specimen PRE12453 (Figure 1), which was part of E.E. Galpin’s personal collection carrying the no. 5967, bears six plant fragments and three labels with relevant information. Firstly, the Galpin’s herbarium label with the original collection data handwritten in ink: “Spergularia/Hanover C.C./Coll: T.R. Sim, Jan.[uary] 1902”. The collector, Dr Thomas R. Sim (1858–1838), was an English botanist specialising in forestry, who worked from September 1894 to September 1902 in the Forestry Service of the Cape Colony as superintendent of plantations, stationed at Fort Cunynghame, just north of Stutterheim (Eastern Cape region), and then he moved to Natal. By that time, he collected abundant plant material in the region that he distributed to contemporary botanists [37], including surely Galpin’s specimen, as deduced from vouchers housed at several South African and European herbaria. Secondly, a label handwritten in pencil and signed by E. Simon reads, “Species mihi adhuc ignota/An Sp. pallida G.Don.??/Ulterius/nomen dabo” (A species still unknown to me. Perhaps *Sp. pallida* G.Don.??/Later I will name it). Finally, a third label with unidentified handwriting in ink reads, “Scrap removed for/Dr. E. Simon,/Vice President of the/Soc. Bot. du Centre-Ouest/France,/(sent through the/S. A. Museum C.T. 5.1.39)”. This latter label is glued exactly in the place left by the fragment forwarded to Dr. Simon via Cape Town. Undoubtedly, PRE12453 is the material from which the name *S. hanoverensis* was invented, and hence, it might be considered as the intended type. This fact is crucial for correctly interpreting and further applying the name in its original concept, as shown below.

In this regard, it is worth mentioning that some confusion exists around the true authorship of *S. hanoverensis*. In IPNI [19] and POWO [13], that species name is connected to “C.Simon”, which is the standard abbreviation of Charles Simon (1908–1987). On the contrary, most of the South African literature as well as TROPICOS [38] attribute it to “Simon”, the standardisation of Eugène Simon (1848–1924), also known as “E.Simon *primus*”, which is partly congruent with the text annotated on the pencil label on voucher PRE12453. However, a detail in that text allows a different but more accurate interpretation, since the fragment missing in that voucher was indeed sent in 1939 to E. Simon, wrganizarwas then the Vice President of the Societé Botanique du Centre-Ouest (France). According to Guédès [39], Eugène Ernest Simon (1871–1967), whose standard form is “E.Simon” (sometimes also cited as “E.Simon *secundus*”, cf. [40]), was the botanist who held the vice presidency of the regional French botanical society since 1908, and who was specifically working on an unfinished monograph on the genus *Spergularia*. Therefore, he was the real proposer of the name *S. hanoverensis*. Irrespectively, the taxon name was never validly published and remained as “nomen nudum”, according to Art. 38 Ex. 1 of the ICN [20].

However, that name is currently available in several checklists and web pages featuring the African and/or South African floras, although no more relevant information has been added besides Nkonki’s [15]. In particular, *S. hanoverensis* was assessed as LC (least concern) by Cholo [41], according to the IUCN red list categories [42], mainly on the basis of the already published information that included mapping of four sites far apart based on vouchers at PRE. Nkonki’s morphological and distributional data [15] are also shown in APD [43] and in GBIF [44].

The extant published information on *S. hanoverensis* is still very scarce and confusing, particularly after reviewing herbarium material of *Spergularia* at PRE. The fragments affixed to voucher PRE12453 (Figure 1) show some outstanding characteristics allowing safe identification:(i)A subshrubby perennial plant, mostly glabrous, with a compact, often many-branched woody underground base;(ii)Stems often prostrate–ascendent;(iii)Leaves somewhat glaucous, with large whitish stipules, very apparent and showy, long-lasting, giving a *Paronychia*-like aspect;(iv)Inflorescence glanduliferous, broadly subdichasial, with inconspicuous bracts;(v)Flowers numerous, small, with white petals;(vi)Capsule small, many seeded;(vii)Seeds small, triangular, blackish-brown, matte, densely papillate, unwinged or with a discolorous, vestigial to well-developed wing.

At first glance, they resemble other South African taxa of *Spergularia*, often misidentified as *S. media*. This might justify the scarcity of references to *S. hanoverensis* in the literature prior to Nkonki [15], and also the neglection of that name in more recent accounts [12]. Other specimens at PRE first identified as *S. hanoverensis* indeed correspond to *S. bocconei* (collection *M.B.Bayer 6006*, PRE762662) or *S. namaquensis* (collection *H. M.Steyn 23*, PRE583500).

Despite the contrasting interpretations of *S. hanoverensis*, the Karoo plants matching Simon’s concept of that species do not fit the current application of the name in recent checklists and databases of the African flora. Further, the unique combination of characters found in Simon’s taxon is missing in the remaining members of the “South African taxa” group of *Spergularia*, as defined by Alonso et al. [8,9]. The newly obtained morphological and molecular evidence support acceptance of the taxon at specific rank.

### 3.2. Taxonomic Treatment

#### ***Spergularia hanoverensis*** E.Simon ex M.Á.Alonso, M.B.Crespo, Mart.-Azorín & Mucina, *sp. nov.*

*Type*: (South Africa. Northern Cape, 3124 (Hanover)). Herbarium-E.E. Galpin, nº 5967—Hanover C(ape). C(olony)., January 1902, *T.R. Sim s.n.* (holotype: PRE12453! Figure 1). *Note*: A fragment of this gathering (an intended isotype) was apparently sent to E. Simon in 1939, but no such material is currently conserved among Simon’s collection at MPU (Montpellier). Maybe it was lost during shipping or as a result of the start of World War II.


*Diagnosis: Planta speciosa ab Spergularia namaquensi et S. glandulosa ob characteribus vegetativis ex parte simillima; a priore caulibus compactis a basi valde lignosis et inflorescentiis terminalibus fere ebracteatis (bracteis minutissimis) accedit, et a posteriore floribus parvis imprimis petalis quam sepalis aequilongis vel parce brevioribus et stylis e basi omnino liberis congruit. Sed plane ab eas distinctissima et bene distinguenda caulibus foliisque glaberrimis; foliis longiore mucronatis (ad 1 mm), stipulis magnis, acuminatis, hyalinis, speciosis, longe persistentibus aspectu Paronychiae; staminibus 7–8; et praecipue seminibus dimorphis, aliis minoribus, 0.5–0.75 × 0.3–0.6 mm, exalatis obovato-cunetatis, et aliis majorib*
*us, 0.4*
*–*
*1.1(*
*–*
*1.4) × 0.6*
*–*
*1.2(*
*–*
*1.4) mm, ala vestigialibus vel perfecta (sed formis aliis in alias tra*
*nseuntes), omnibus disco densiore et uniformiter tuberculato-papilloso.*


*Description*: *Subshrub*, mostly glabrous excepting the glanduliferous inflorescence, with a compact, often many-branched woody underground base. *Stems* up to 30 cm high, but commonly smaller, usually prostrate, subcespitose, slightly nodose, with ascending branches. *Leaves* 4–10(–17) × 0.5–0.7 mm, narrowly linear, semicylindrical and subcanaliculate, green to slightly glaucous-green, caducous when withering, ending in a whitish to yellowish mucro up to 1 mm long; *stipules* 4–6 × 1–2 mm, whitish-scarious, glabrous, triangular-acuminate, showy and apparent, often reaching at least half the leaf length (namely in the axillary leaf fascicles, where stipules are longer than leaves themselves), those on the young shoots fused up to the basal third and finally only at the base, long-lasting after leaf abscission, and giving a *Paronychia*-like aspect. *Inflorescence* a subdichasial cyme, broadly branched, many flowered (5–7 flowers per branch), and densely covered with glanduliferous short hairs; *bracts* 1–2.5 mm long, slightly longer than stipules, inconspicuous, much shorter than leaves. *Flowers* pentamerous, on erect-patent to patent slender pedicels 2.5–5 mm long at anthesis. *Sepals* 2–3.3 × 1–1.5 mm, slightly accrescent in fruit, oblong to elliptic, obtuse to subacute, with a central green band 0.5–0.7 mm broad, and scarious margins 0.3–0.4 mm broad (wider in the inner sepals), patent to slightly deflexed at anthesis. *Petals* 1.8–2.8 × 1–1.2 mm, about equalling to slightly shorter than sepals, white, elliptic, entire. *Stamens* 7–8, slightly shorter than petals, filament up to 3 mm long, filiform, slightly widened at the base, whitish, anther 0.3–0.4 mm long, yellow, dorsifixed. *Ovary* ca. 1–2 mm, subglobose, yellowish; styles 3, ca. 1 mm long, free from the base, yellowish, with short apical stigmata. *Capsule* 3–4 × 2.5–3.5 mm, broadly ovate, slightly longer than sepals, glabrous, shining, yellowish-green outside but reddish inside, opening by three slightly recurved valves, on patent to reflexed pedicels 4–10 mm long, up to 3 times longer than sepals. *Seeds* numerous, dimorphic; some of them unwinged, 0.5–0.75 × 0.3–0.6 mm, ovate-cuneate to subtriangular in outline, flattened, blackish-brown and matte, the others similar but larger, 0.7–1.1(–1.4) × 0.6–1.2(–1.4) mm, with disk 0.6–0.9 × 0.4–0.8 mm and a discolorous (whitish to greyish), vestigial to entirely developed, eroded wing (both types usually present in a single capsule, with intermediate stages); *testa* ornamented in all cases with minute irregular tubercles and densely covered all over with stalked globose and also minutely tuberculated papillae.

*Etymology*: The specific epithet (*hanoverensis*, −*e*) refers to Hanover, a small town in the Karoo region of the Northern Cape province in central South Africa, where the plant was collected and is native to. Provided that E.E. Simon was the first to recognise this taxon as new “in schedis”, we preserve the original name he chose later.

*Phenology*: Flowering in late October–early January (occasionally in July–August), fruiting in November–February (occasionally in August–September).

*Habitat and distribution*: *Spergularia hanoverensis* is an edaphic specialist species usually found on seasonal stream banks, in riverbeds and depressions, often on saline calcareous-clayish or sandy soils but sometimes among rocks or stony saline substrates (Figure 2). The elevation of the localities ranges between 700 and 1400 m. The known distribution of the species extends through most of the southern part of the Karoo region in central and western South Africa, ranging from Ceres and Calitzdorp in the Western Cape to Hanover in the Northern Cape province (Figure 3). That territory is mostly included in the Nama-Karoo (NK) biome and reaches the southern Succulent-Karoo biomes (mostly the SKk, SKt, and SKv bioregions) sensu Mucina and Rutherford [16], where it occurs in the so-called “Bushmanland vloere” (code AZi 5). In those areas, the climate is subdesert arid and continental (only scarcely ameliorated by the ocean influences), with average temperatures ranging from −5 °C in winter to 43 °C in summer and frosts being usual at high elevations. The average annual precipitation varies between 100 and 500 mm, though rather differently distributed, with the rainfall occurring mostly during late summer (December to April) with a peak in March [16].

*Notes*: Wild populations of *S. hanoverensis* include numerous individuals covering a large territory in South Africa, and no special threats are known so far that might lead to any inferred decline in either the number of populations or the number of individuals. Therefore, its conservation status is suggested here as least concern (LC) according to IUCN [42]. Nonetheless, extensive fieldwork is still needed to locate new populations in suitable habitats among the known populations, which will allow completing the distribution area of *S. hanoverensis* or/and detecting eventual variation of adaptative characteristics in distinct environments. The identification key in Appendix A can help to do this.

*Other studied materials*: South Africa. **Northern Cape province**: 3020 (Brandvlei): between Brandvlei and Williston, Karreekop farm (−DC), 30°58′59″ S, 20°39′18″ E, 987 m, P36, 24 August 2022, *M.Mart.Azorín, M.B.Crespo, M.Á.Alonso, J.L.Villar & M.Pinter* (ABH83276). 3119 (Calvinia): Calvinia C(ape). P(rovince). (−BD), 1936, ut *Spergularia marginata* Kit., *A.A.Schmidt 408* (PRE0405924). 3119 (Calvinia): banks of Kareehoutrivier, 24 km south of Bo-Downes (−DD), cushion along the seasonal stream banks, 20 November 1983, *D.Snijman 772* (NBG). 3120 (Williston): Calvinia district, Rietfontein, on road to Brandvlei (−AC), karoo, sandy soil, 29 November 1986, *G.Germishuizen 4022* (PRE0694503). 3121 (Fraseburgh): Williston District, Farm Grootfontein, about 42 km north Williston, on road to Rheebokhyer, about 7 km west of farmhouse (−AA), elev. 1207 m, karoo, among rocks, alongside dry, sandy riverbed, 2 April 1993, *G.Germishuizen 6524* (PRE0791338). 3124 (Hanover): outskirts of Hanover, on S. side (−AB), in white calcareous clay in vley, locally freq., elev. 4500 ft, 27 February 1956, *J.P.H.Acocks 18797* (3 sheets) (PRE0405795). 3124 (Hanover): SW side of Hanover (−AB), kalk vlei, elev. 4500 ft, 19 February 1959, *J.P.H.Acocks 20246* (3 sheets) (PRE0405797). 3220 (Sutherland): Roggeveld, Soekop, Onderste grasvlakte camp (−AA), 32°03′24.5″ S, 20°08′39.5″ E, elev. 1193 m, 27 September 2006, *H.Rosch 656* (NBG209250). 3220 (Sutherland): SW of Middelpos, on the road to Ganaga Pass, Zoekop Farm, Witdam (−AA), 32°03’16.5″ S, 20°08’39.2″ E, elev. 1190 m, 25 August 2022, *M.Mart.Azorín, M.B.Crespo, M.Á.Alonso, J.L.Villar & M.Pinter* (ABH83287; ABH83288). **Western Cape province**: 3219 (Wuppertal): Swartruggens, Knolfontein (−DC), 32°51′17.2″ S, 19°36′03″ E, elev. 1190 m, 3 February 2011, *I.Jardine 1522* (NBG276922). 3219 (Wuppertal): Swartruggens, Knolfontein −DC), 32°51′17.2″ S, 19°37′05″ E, elev. 1186 m, 12 December 2011, *I.Jardine 1760* (NBG277559). 3319 (Worcester): Ceres, Bokkerivier Farm (−BD), 11 November 1963, *L.J.Booysen 111* (NBG).

### 3.3. Taxonomic and Phylogenetic Relationships of Spergularia hanoverensis

In the context of an ongoing phylogenetic survey on the South African taxa of *Spergularia* based on *trnL-F* cpDNA and ITS2 nDNA sequences, we recently reported [8,9] the first preliminary phylogenetic trees that included South African taxa of that genus. Our results were congruent with the plastid phylogeny of *Sperguleae* (sensu [3]) obtained by Kool and Thulin [7], and also supported that *trnL-F* and ITS2 regions offer information useful enough for credible phylogenetic reconstructions of *Spergularia*.

Adding samples of *S. hanoverensis* to our molecular matrix yields trees almost identical in general topology to those obtained in our previous contributions [8,9]. Further, the new species falls nested in the “South African taxa” subclade of Clade A (sensu [8,9]) in both the plastid and the nuclear phylogenies (Figure 4 and Figure 5), as recovered in our BI consensus trees (in which PP values are placed above branches and BP values below branches, respectively, from the ML and MP analyses). However, the internal relationships among the four members of that southern lineage are not equally resolved, often with low PP and BP values.

The aligned *trnL-F* database was 990 bp, 147 of which (14.85%) were potentially parsimony informative. Analyses of this dataset using NJ, MP, ML, and BI methods yielded trees with similar topologies and similar bootstrap and branch length values. The obtained *trnL-F* BI phylogenetic tree (Figure 4) recovers all four South African taxa in the strongly supported Clade A (1.00 PP, 97/100 BP), in which both samples of *S. hanoverensis* are unresolved together with the remaining accessions of *S. glandulosa*, *S. namaquensis,* and *S. quartzicola* to form the group of “South African taxa” (1.00 PP, 87/100 BP). They all fall together in a polytomic group (1.00 PP, 86/100 BP) with mostly Southern Hemisphere taxa, namely *S. villosa* (Pers.) Cambess., *S. tasmanica* (Kindb.) L.G.Adams, *S. ramosa* Cambess., and *S. pissisii* I.M.Johnst. They all are sister to the well-supported subclade (0.81 PP, 54/100 BP) formed by *S. denticulata* (Phil.) Phil. and the Pacific North American *S. macrotheca* (Hornem. ex Cham. & Schltdl.) Heynh.

The aligned ITS2 database was 282 bp, 41 of which (14.54%) were potentially parsimony informative. Further, analyses using NJ, MP, ML, and BI methods revealed trees with similar topologies and similar bootstrap and branch length values. The obtained ITS2 BI phylogenetic tree (Figure 5) recovers both samples of *S. hanoverensis* as well-supported sister (0.99 PP, 88/94 BP) of the well-supported subclade of *S. glandulosa* (0.93 PP, 80/79 BP), a subshrub from saline coastal areas of southwestern and southern South Africa. They both are well-supported sister (0.99 PP, 88/94 BP) to the subclade including the Namaqualand woody subshrubs *S. quartzicola* and *S. namaquensis*, which constitute a strongly supported group (1.00 PP, 96/97 BP) not well resolved internally. All these are strongly supported sister (1.00 PP, 93/95 BP) to a third subclade formed by the reddish-flowered *S. rubra* and *S. rupicola* (1.00 PP, 88/92 BP) plus the Central European *S. echinosperma* (Čelak.) Asch. & Graebn. All those constitute the strongly supported (0.98 PP, 90/97 BP) Clade A.

Those molecular relationships of *Spergularia hanoverensis* to the remaining members of the “South African taxa” group correlate to some morphological traits obtained from our morphological studies. Table 3 shows the most important characters of *S. hanoverensis* compared to other related perennial relatives of the “South African taxa” group, illustrating the intraspecific variation of the newly described species. *Spergularia hanoverensis* shares some floral resemblance with *S. glandulosa*, such as the small white flowers with petals equalling sepals in length, and the styles entirely free from the base, features that might justify some phylogenetic closeness as recovered in our ITS2 tree (Figure 5). However, *S. hanoverensis* shows a compact habit; leaves glabrous and long acuminate (ca. 1 mm), bearing at base long-lasting, broad stipules; and inflorescences multiflowered, subdichasial, with minute bracts. These characters all allow easy separation at first glance. In fact, *S. hanoverensis* exhibits a combination of characters missing in any known members of the South African group. However, some of them occur in any of the three other taxa. In particular, the compact and usually many-branched woody underground base and the multiflowered subdichasial inflorescences with minute bracts are shared with *S. namaquensis*, and the entirely glabrous and broadly stipulate leaves are also present in *S. quartzicola*. Nonetheless, both latter taxa show larger flowers with 10 stamens and columnar, long-fused styles, instead of the smaller flowers with 7–8 stamens and free styles of *S. hanoverensis*. The main morphological vegetative features exhibited by *S. hanoverensis* inform about some adaptative characteristics of the species to high environmental stress (i.e., high salinity and persistent dry soil at high-elevation open habitats), such as the uniformly subshrubby compact habit, along with the slightly glaucous colour of leaves and the long-lasting conspicuous white-hyaline stipules (bringing a silvery *Paronychia*-like appearance at anthesis) that favour sunlight reflection.

Finally, the capsule and seed features offer the most remarkable diagnostic differences among all those South African species. *Spergularia hanoverensis* produces the smallest capsules (ca. 3–4 × 2.5–3.5 mm) and also smaller seeds (0.4–0.5 mm long). These are markedly dimorphic, some of them being unwinged, ovate-cuneate to subtriangular in outline, flattened, blackish-brown and matte, whereas the others are similar but with a discolorous (whitish to greyish), vestigial to entirely developed, eroded wing, and testa surface ornamented with minute irregular tubercles and densely covered all over with stalked globose and also minutely tuberculated papillae. Both seed types are sometimes present in a single capsule (Figure 6a–c,f,g), with intermediate stages among individuals (Figure 6b) and among populations. This fact makes *S. hanoverensis* the only known member of the “South African taxa” group with such dimorphic seeds.

Seed dimorphism is rather frequent in *Spergularia* [40,45,46,47,48,49,50], and for a long time, it has been considered a good diagnostic character for species delimitation [51]. As occurring in other dimorphic taxa such as the widespread annual pinkish-flowered *S. marina* (L.) Besser (incl. *S. salina* J. & C. Presl) [52,53], heteromorphism also includes variable seed morphology as well as considerable variation in seed size in *S. hanoverensis*, the winged seeds being larger, and their disk is also larger than the unwinged seeds. However, seeds of *S. hanoverensis*, although dimorphic in size and gross morphology (winged and unwinged), are homogeneous in micromorphology, with traits quite distinct from *S. marina* [54]. According to our observations from herbarium material, no correlation appears between seed features and their geographical provenance. Hence, individuals from different sites apparently exhibit seeds with similar size and gross morphology variation patterns. In other words, most of the variation in seed size occurs within rather than between populations, as also reported for *S. marina* [53]. 

Species with seed dimorphism combine different morphological syndromes for a more efficient dispersal in contrasting habitats, the wind being the vector for winged seeds and the water for unwinged ones, as suggested by Telenius and Torstensson [55]. Acquisition of dimorphism in *S. hanoverensis* might have brought a positive evolutionary effect for dispersal in the inland karroid saline ecosystems it inhabits in South Africa (i.e., seasonally wet depressions, riverbeds and ravines with seasonal flow, etc.), by combining anemochory (winged seeds) and hydrochory (unwinged seeds). Its phylogenetically close relatives from Namaqualand (*S. namaquensis* and *S. quartzicola*) also occur in subdesert karroid ecosystems, mostly in dry sandy or quartz substrates. Therefore, the production of unwinged seeds would have no favourable adaptative value in sites not necessarily connected to marsh habitats.

## 4. Conclusions

Although *Spergularia hanoverensis* has been widely interpreted in a sense that includes several biological entities, the name is validated and circumscribed here as it was first outlined “in schedis” by E.E. Simon. Our molecular and morphological data are distinctive enough to accept it as a proper species, which falls together with taxa in the “South African taxa” group of *Spergularia*. It is a halophytic specialist, mostly endemic to the inland saltmarsh habitats of the Nama-Karoo biome (central-western South Africa). It shows some adaptative characteristics to stressful environments with high salinity and dry soils at high-elevation sites.

New data from wild populations are needed to complete the distribution area of *S. hanoverensis* and to test the accuracy of our observations on herbarium vouchers regarding within- or between-population variation in seed size and gross morphology, as well as eventual variation of adaptative characteristics of the species to different environments. The dichotomous identification key in Appendix A will surely help to that purpose.

## Figures and Tables

**Figure 1 plants-12-02481-f001:**
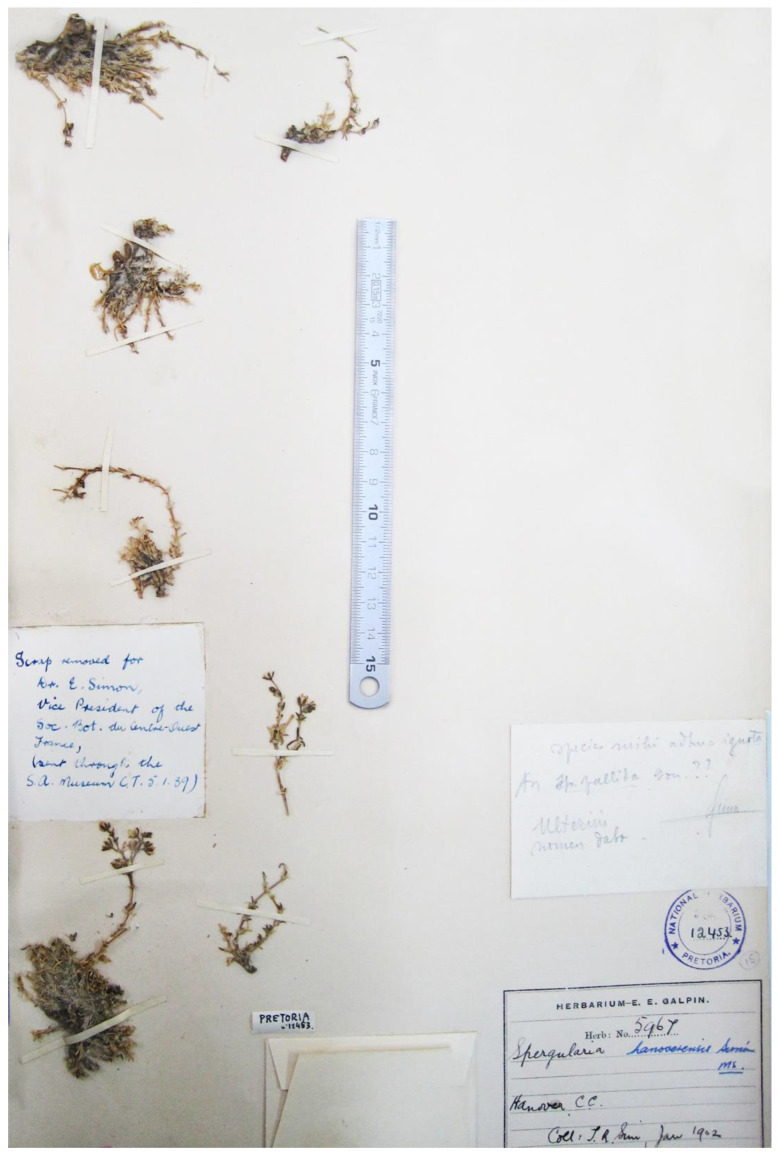
Voucher designated here as holotype of *Spergularia hanoverensis* (PRE12453), which was studied and annotated by E.E. Simon as belonging to an unknown species (^©^SANBI, Herbarium, Pretoria). Ruler length = 15 cm (minimum scale = 0.5 mm).

**Figure 2 plants-12-02481-f002:**
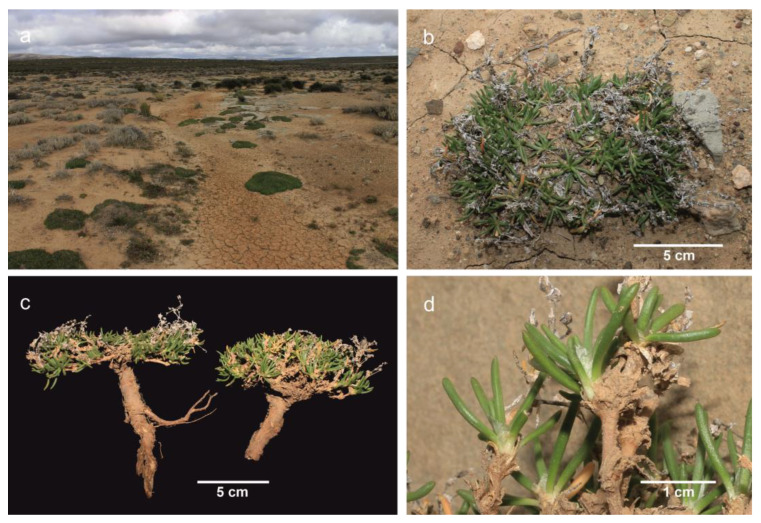
*Spergularia hanoverensis* near Zoekop Farm, SW of Middelpos, on the road to Ganaga Pass (Northern Cape province). (**a**) General view of habitat; (**b**) close-up image of plants in habitat; (**c**) plant habit with root system and withered inflorescences; (**d**) detail of leaves and stipules.

**Figure 3 plants-12-02481-f003:**
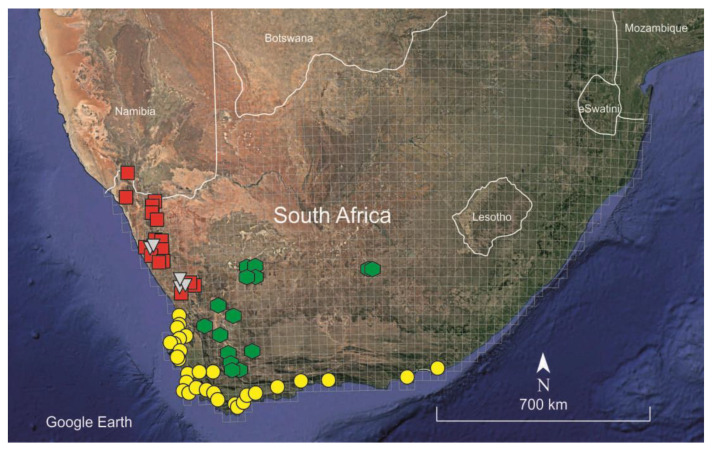
Distribution map of the known species of the “South African taxa” group of *Spergularia*: *S. hanoverensis* (green hexagons); *S. glandulosa* (yellow circles); *S. namaquensis* (red squares); and *S. quartzicola* (greyish triangles).

**Figure 4 plants-12-02481-f004:**
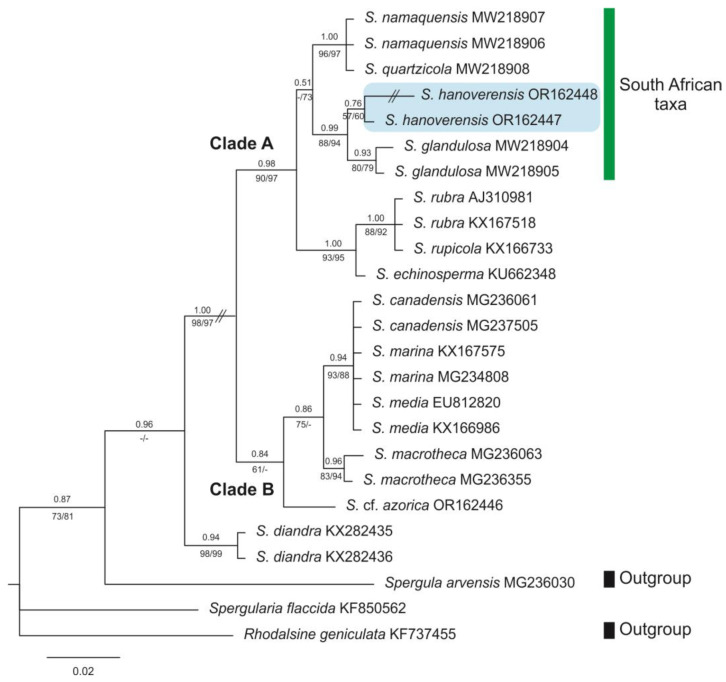
Bayesian phylogenetic tree of *Spergularia* accessions from *trn*L–*trn*F plastid DNA sequences. The position of the South African taxa is marked in blue in Clade A, where *S. hanoverensis* is not fully resolved. Numbers above branches indicate posterior probabilities (PP), whereas numbers below branches correspond to bootstrap percentages (BP) from both ML/MP (strict consensus) trees.

**Figure 5 plants-12-02481-f005:**
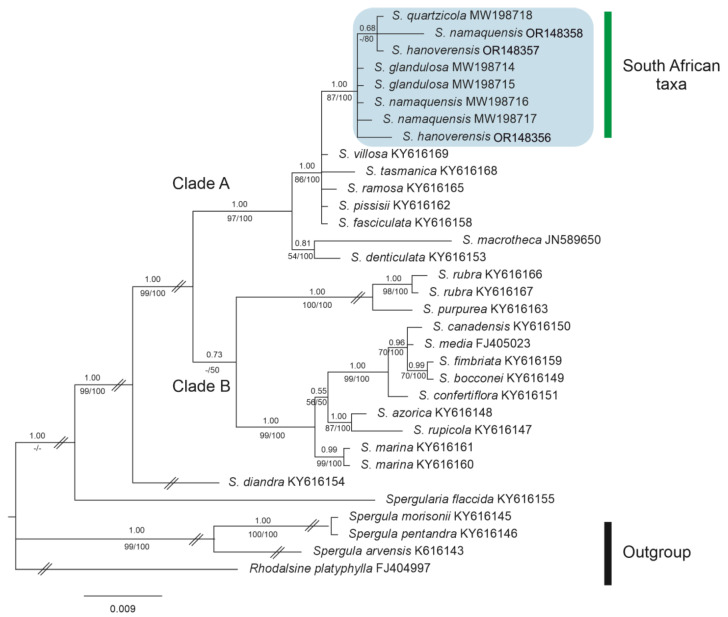
Bayesian phylogenetic tree of *Spergularia* accessions from 5.8S-ITS2 nuclear DNA sequences. The position of the South African taxa is marked in blue in Clade A. Numbers above branches indicate posterior probabilities (PP) from the Bayesian analysis, whereas numbers below branches correspond to bootstrap percentages (BP) from both ML/MP (strict consensus) trees.

**Figure 6 plants-12-02481-f006:**
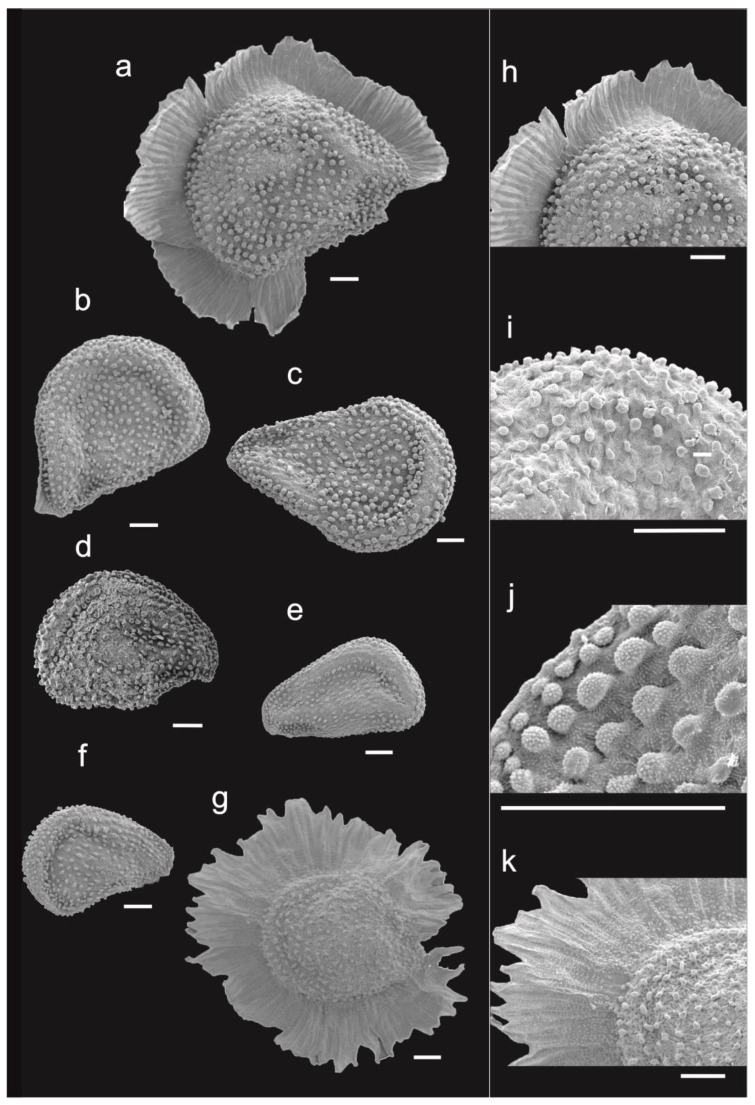
Seed morphology variation in *Spergularia hanoverensis*. (**a**–**c**) Northern Cape: Rietfontein: large dimorphic seeds from a single capsule (PRE0694503); (**d**) Northern Cape: Williston, Farm Grootfontein: large unwinged seed (PRE0791338); (**e**) Northern Cape: Hanover: small unwinged seed (PRE0405795); (**f**,**g**) Northern Cape: Calvinia: small dimorphic seeds from a single capsule (PRE0405924); (**h**) Rietfontein: testa ornamentation of winged seed (PRE0694503); (**i**) Northern Cape: Calvinia: testa ornamentation of unwinged seed (PRE0405924); (**j**) Northern Cape: Hanover: detail of testa papillae (PRE0405795); (**k**) Northern Cape: Calvinia: testa ornamentation of winged seed (PRE0405924). Scale bars = 100 μm.

**Table 1 plants-12-02481-t001:** Seed samples of *Spergularia hanoverensis* for SEM studies, with provenance and vouchers.

Locality	Voucher
ZA: Northern Cape, Rietfontein to Brandvlei, *G.Germishuizen 4022*	PRE0694503
ZA: Northern Cape, Farm Grootfontein, *G.Germishuizen 6524*	PRE0791338
ZA: Northern Cape, Calvinia, *A.A.Schmidt 408*	PRE0405924
ZA: Northern Cape, Hanover, *J.P.H.Acocks 18797*	PRE0405795

**Table 2 plants-12-02481-t002:** List of *Spergularia* accessions generated specifically for this study. See Alonso et al. [8] for additional details on the rest of accessions used in the analyses.

Taxon	Provenance (Herbarium Voucher)	Source	GenBank Accession Code	
			*trnL*-*trnF*	5.8S-ITS2
*S.* cf. *azorica* (Kindb.) Lebel	Spain: Tenerife, Pto, de la Cruz (ABH79973)	This paper	–	OR162446
*S. hanoverensis* E.Simon ex M.Á.Alonso et al.	South Africa: Zoekop Farm (ABH83288)	This paper	OR148356	OR162447
	South Africa: Karreekop Farm (ABH83276)	This paper	OR148357	OR162448
*S. namaquensis* Schltr. Ex M.Á.Alonso et al.	South Africa: Skoverfontein (ABH83197)	This paper	OR148358	–

**Table 3 plants-12-02481-t003:** Comparison of morphological characters of *Spergularia hanoverensis* with its perennial relatives of the “South African taxa” group.

	*S. glandulosa*	*S. namaquensis*	*S. media*	*S. quartzicola*	*S. hanoverensis*
Height (cm)	up to 15	up to 20	5–50(–65)	up to 30	up to 30
Habit	subshrub	subshrub	perennial herbs, weakly lignified at base	subshrub	subshrub
Stems	caespitose, strongly nodose, procumbent to prostrate	erect, nodose, with ascending branches	diffuse, procumbent to prostrate	erect, nodose below, with subfastigiate branches	compact, slightly nodose, with ascending branches
Young branches indumentum	densely covered all over with gladuliferous, pluricellular hairs	densely covered all over with glanduliferous, pluricellular hairs	glabrous	glabrous (occasionally sparsely glanduliferous when young)	glabrous
Leaves (mm)	8–20 × 0.5–1	8–25 × 0.8–1.2	10–35 × 0.3–2.5	8–45 × 0.5–1.2	4–10(–1.7) × 0.5–0.7
Indumentum of leaves	covered with glanduliferous hairs, occasionally glabrescent	covered with glanduliferous hairs, occasionally glabrescent	glabrous	glabrous	glabrous
Leaf mucro (mm)	0.2–0.4	0.4–0.5	0–0.2	0.2–0.5	0.5–1.0
Stipules (mm)	3–6 × 3–4	2–4 × 2–4	2.3–2.5 × 1.3–2	3–5 × 2–2.6	4–6 × 1–2
Stipules shape and connation	triangular to broadly triangular, those of young stems fused up to 1/3 of their length	triangular to broadly triangular, those of young stems fused up to 1/3 of their length	triangular, not acuminate, those of young stems fused up to 1/2 of their length	narrowly triangular, long acuminate, those of young stems fused up to 1/3 of their length	triangular-acuminate, those of young stems fused up to 1/3 of their length
Stipules colour, texture, and indumentum	whitish-scarious, with glanduliferous hairs in the basal part	whitish-scarious, with glanduliferous hairs in the basal part	whitish-scarious, glabrous	whitish-scarious, glabrous	whitish-scarious, glabrous
Floral bracts/stipules	bracts longer to slightly longer than stipules	bracts slightly longer than stipules	bracts slightly longer than stipules	bracts longer than stipules	bracts longer than stipules
Inflorescence	monochasial to dichasial cyme, narrowly branched	dichasial cyme, broadly branched	dichasial cyme, broadly branched	dichasial cyme, broadly branched, rarely monochasial	dichasial cyme, broadly branched
Bract length (mm)	(2–)6–10	(2–)5–9	1.7–2	(1.5–)3–7	(1–)1.5–2.5
Indumentum of bracts	glandular-pubescent	glandular-pubescent	glabrescent to glandular-pubescent	glandular-pubescent	glandular-pubescent
Sepals (mm)	2–6 × 2–3, with narrowly scarious margins, erect-patent at anthesis	5–7.5 × 4–5, with broadly scarious margins, patent to slightly deflexed at anthesis	4–6 × 2, with broadly scarious margins, erect-patent at anthesis	4.6–6 × 1.5–3, with narrowly scarious margins, strongly deflexed at anthesis	2–3 × 1–1.5 with narrowly scarious margins, patent to slightly deflexed at anthesis
Indumentum of sepals	glandular-pubescent	glandular-pubescent	glandular-pubescent	glandular-pubescent	glandular-pubescent
Petal colour	white	white	white or sometimes pink at apex	white	white
Sepals/petals	slightly shorter than to equalling sepals	up to 1.5 times longer than sepals	up to 1.2 times longer than sepals	up to 1.5 times longer than sepals	slightly shorter than to equalling sepals
Petals (mm)	2–5 × 1.5–3	6–9 × 4–5	4–5 × 1.8–2	6–8 × 4–5	1.8–2.8 × 1–1.2
Stamens	10, widened at base	10, not widened at base	(7–9)10, widened at base	10, dissimilar (5 widened at base)	7–8, slightly widened at base
Stamens/petals	shorter than petals	shorter than petals	shorter than petals	shorter than petals	shorter than petals
Anther	0.5–0.8	0.6–1	0.9–1	0.8–1	0.3–0.4
Styles	3, free from base	3, fused in column to half or beyond	3, free from base	3, fused in column to half or beyond	3, free from base
Style length (mm)	0.5–0.7	1–1.1	0.5–0.6	1.5–2	1
Capsule (mm)	4–6.5 × 1.5–3.5	6–9 × 3–4	(5–)6–9 × (3.5–)4–6	5.5–6.5 × 3–3.5	3–4 × 2.5–3.5
Capsule/sepals	slightly longer than sepals	equalling to slightly longer than sepals	1/3 longer than sepals	equalling to slightly longer than sepals	slightly longer than sepals
Seed	winged	winged	winged	winged, discolour	winged, discolour, to unwinged
Seed colour	discolour, scarious whitish wing and dull brown to blackish disk	discolour, scarious whitish wing and blackish disk	discolour, scarious whitish wing and blackish disk	discolour, scarious whitish wing and blackish disk	blackish-brown and matte, or discolour, with scarious whitish wing
Size of seed (mm)	0.9–1.3 × 1.2–1.3	1.1–1.3 × 1.2–1.4	1.2–1.5 × 1.3–1.6	1.1–1.2 × 1.1–1.2	0.5–0.75 × 0.3–0.6 and 0.7–1.1(–1.4) × 0.6–1.2(–1.4)
Seed disk (mm)	0.6–0.7 × 0.6–0.9, loosely covered with conical tubercles	0.6–0.7 × 0.6–0.8, smooth or with conical tubercles only on edges	0.6–0.8 × 0.5–0.7, smooth	0.6–0.7 × 0.6–0.7, smooth or with granular protuberances, mostly on edges	0.5–0.9 × 0.4–0.9, densely covered with globose, stalked papillae
Width of seed wing (mm)	0.3–0.4	0.4–0.6	0.3–0.5	0.5–0.6	0–0.5
Wing edge	entire to slightly eroded	entire to slightly eroded	entire to slightly eroded	deeply and irregularly lacerate	eroded, when present

## Data Availability

DNA sequence data generated in the present research are available at GenBank (https://www.ncbi.nlm.nih.gov/genbank/).

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
