# Peer review of "Spergularia hanoverensis (Caryophyllaceae): Validation and Recircumscription of a Misinterpreted Species from South Africa"

_plants, 2023, doi:10.3390/plants12132481_

Round 1

Reviewer 1 Report

General comment:

This paper studied "Spergularia hanoverensis Simon" distributed in the semi-desert ecosystem of the central and western interior of South Africa, and found that its previous classification was wrong, so as to re-establish the complete morphological description, habitat, distribution and phylogenetic data of this plant, and correct the previous error, which is of great significance.

Major comment:

1.      This work is very detailed and meaningful in correcting a previous error in plant taxonomy by finding herbariums and some personal collections, but I think it would be more complete if this paper could increase the morphology and adaptive characteristics of the extant species (Spergularia hanoverensis) in different environments through extensive sampling. At least add some discussion about it

2.      There is too much scrutiny of historical data in the outcome of the article, and these studies do not seem to be particularly strongly supported by the literature, perhaps the conclusions of this paper should be based more on the results of phylogenetic analysis and less on the reliability of previous data. At least add some discussion about it

Minor comment:

1.      In line 51, "and references therein" seems redundant

2.      Line 135, where "for" and possibly "at" in context?

3.      The source of the samples in 3.1 can be placed in the first part of the article, and the result only introduces the results of these samples

4.      In line 266, perhaps it should be changed to "no more relevant information", which obviously could not have been updated many years ago, and should be a paper record. So it's better to use "more" information, because maybe somewhere else there are other additions

Reviewer 2 Report

Line 122. 'mostly kept'. What are the exceptions?

Line 217. 'should' on what basis?

Reviewer 3 Report

I have read with interest this manuscript. I normally appreciate when an integrative taxonomy approach is properly followed, thus depicting the status of a taxon on the basis of morphology, molecular phylogeny, and ecology. Thus, in my humble opinion, this manuscript should be welcome on Plants.

However, I have two major issues, thus my request for a major revision.

The first issue is related to the morphological analysis. The authors state in lines 99-101 that morphology was investigated in both fresh and preserved specimens. However, they do not specify anywhere how many specimens for each taxon were investigated. The rich table 3, in which they depict the morphological features of the 5 African taxa of the genus, thus, is quite difficult to interpreter. If the authors investigated several specimens per taxon from several populations, thus accounting for intra-specific variability, then the table can be used for supporting the results of molecular phylogeny (at lest as far as ITS is concerned).

My second concern is mostly related to the number of samples which were used for molecular phylogeny. As far as I can understand, DNA was extracted from the herbarium specimens only. The phylogenetic tree, at least as far as ITS is concerned, supports the delimitation of S. hanoverensis. However, given that the authors have fresh material collected in the field, material which was used for morphological investigations, why they did not used this material as well for molecular analysis? Having more than 2 specimens could better support the delimitation of the taxon.

As a further note, table 2 is missing the GenBank accession codes.

Best regards.

In general, the manuscript is mostly well written, but I found that some parts, and the abstract in particular, require a revision by a native English speaker.

Round 2

Reviewer 1 Report

I have no further comments.

Author Response

Dear Reviewer 1,

Thanks very much for your previous comments and help. 

Best wishes

Reviewer 3 Report

Dear authors, 

I appreciate you effort in improving the manuscript.

I have only a minor suggestion, thus the minor revision. Since you wrote in the response to my observations that adding further samples in the phylogenetic trees did not change the outcome, you should specify the same also while discussing your results in the manuscript.

Best regards

SM

I would suggest a further check, even if it is already quite good, since several changes have been done in the manuscript.

Author Response

Dear Reviewer 3,

Many thanks indeed for your comments, and also the final minor suggestion. I have added a short sentence to the previously added comment in Mat & Met, as follows: "For Spergularia hanoverensis, sampling from herbarium material was not permitted, and hence only silicagel dried material from two wild populations (one sample per population) was utilised. Addition of further samples from those same populations did not modify the phylogenetic trees."

I hope it is fine now. Your contribution much improved the final text, so thanks again.

Cheers,